**African Smoke Particles Act as Cloud Condensation Nuclei in the Wintertime Tropical North Atlantic Boundary Layer over Barbados**

Haley M. Royer[1], Mira L. Pöhlker[2,3,4*], Ovid Krüger[2], Edmund Blades[5,6], Peter Sealy[5], Nurun Nahar Lata[7], Zezhen Cheng[7], Swarup China[7], Andrew P. Ault[8], Patricia K. Quinn[9], Paquita Zuidema[1], Christopher Pöhlker[2], Ulrich Pöschl[2], Meinrat Andreae[2,10,11] and Cassandra J. Gaston[1*]

[1]Department of Atmospheric Sciences, Rosenstiel School of Marine, Atmospheric, and Earth Science, University of Miami, Miami, FL, United States of America

[2]Department of Multiphase Chemistry, Max Planck Institute for Chemistry, Mainz, Germany

[3]Leipzig Institute for Meteorology, Leipzig University, Leipzig, Germany

[4]Experimental Aerosol and Cloud Microphysics Department, Leibniz Institute for Tropospheric Research, Leipzig, Germany

[5]Barbados Atmospheric Chemistry Observatory, Ragged Point, Barbados

[6]Queen Elizabeth Hospital Barbados, Bridgetown, Barbados

[7]Environmental Molecular Sciences Laboratory, Pacific Northwest National Laboratory, Richland, WA, United States of America

[8]Department of Chemistry, University of Michigan, Ann Arbor, MI, United States of America

[9]Pacific Marine Environmental Laboratory, National Oceanic and Atmospheric Administration, Seattle, WA, United States of America

[10]Department of Geology and Geophysics, King Saud University, Riyadh, Saudi Arabia

[11]Scripps Institution of Oceanography, University of California San Diego, La Jolla, California,
United States of America
*Corresponding Authors:
Mira L. Pöhlker: Email: poehlker@tropos.de, Phone: +49 6131 305 7020
Cassandra J. Gaston: Email: cgaston@.miami.edu, Phone: (305)-421-4979

**Abstract**

The number concentration and properties of aerosol particles serving as cloud condensation nuclei (CCN) are important for understanding cloud properties, including in the tropical Atlantic marine boundary layer (MBL) where marine cumulus clouds reflect incoming solar radiation and obscure the low-albedo ocean surface. Studies linking aerosol source, composition, and water uptake properties in this region have been conducted primarily during the summertime dust transport season, despite the region receiving a variety of aerosol particle types throughout the year. In this study, we compare size-resolved aerosol chemical composition data to the hygroscopicity parameter $\kappa$ derived from size-resolved CCN measurements made during the EUREC[4]A and ATOMIC campaigns from January to February 2020. We observed unexpected periods of wintertime long-range transport of African smoke and dust to Barbados. During these periods, the accumulation mode aerosol particle and CCN number concentrations as well as the proportions of dust and smoke particles increased, whereas the average $\kappa$ slightly decreased ($\kappa = 0.46 \pm 0.10$) from marine background conditions ($\kappa = 0.52 \pm 0.09$) when the particles were mostly composed of marine organics and sulfate. Size-resolved chemical analysis shows that smoke particles were the major contributor to the accumulation mode during long-range transport events, indicating that smoke is mainly responsible for the observed increase in CCN number concentrations. Earlier studies conducted at Barbados have mostly focused on the role of dust on CCN, but our results show that aerosol hygroscopicity and CCN number concentrations during wintertime long-range transport events over the tropical North Atlantic are also affected by African smoke. Our findings highlight the importance of African smoke for atmospheric processes and cloud formation over the Caribbean.

**Introduction**

Aerosol particle number, size, hygroscopicity, and chemical mixing state determine cloud droplet formation and, thus, fundamentally affect the radiative properties and lifetime of clouds (Albrecht, 1989; McFiggans et al., 2006; Quinn et al., 2008; Twomey, 1977; Zuidema et al., 2008). Quantifying the effect of aerosols on cloud radiative forcing, however, is still the single largest source of uncertainty in predicting temperature increases associated with climate change (Forster et al., 2021). This uncertainty is especially important to resolve in marine regions where aerosol-cloud interactions are understudied, even though the majority of Earth's surface is covered by oceans (Carslaw et al., 2013). The existing literature that explores marine aerosol-cloud interactions does so primarily in the mid to high latitudes of the North Atlantic with few studies focusing in tropical latitudes where shallow cumulus clouds form (Allan et al., 2008; Behrenfeld et al., 2019; Klingebiel et al., 2019; Rauber et al., 2007; Sorooshian et al., 2020). Shallow cumulus clouds are important for Earth's climate as they are one of the most geographically pervasive cloud types and can influence Earth's radiative budget by reflecting incoming radiation over the low-albedo ocean surface.

Aerosol research conducted in the tropical Atlantic has focused largely on the long-range transport of mineral dust from North Africa in the summertime. Long-range African dust transport occurs when emitted desert dust is lofted above the marine boundary layer (MBL) into the Saharan Air Layer (SAL) and is propagated westward (Carlson & Prospero, 1972). As dust is transported westward, it can mix into the underlying moist MBL and deposit into the Atlantic Ocean and Caribbean Sea as well as Western Atlantic land masses such as South America, the Caribbean islands, and North America (Barkley et al., 2019; Carlson & Prospero, 1972; Prospero et al., 1981, 2020). Some studies have attempted to understand the effects of long-range transported

dust on cloud droplet formation and water uptake with varying results depending on the degree
of aging that dust experiences during transport (Allan et al., 2008; Denjean et al., 2015;
Kristensen et al., 2016; Rosenfeld et al., 2001; Wex et al., 2016). However, these studies provide
conflicting results on whether dust particles are hygroscopic and numerous enough to
appreciably impact CCN concentrations in the tropical Atlantic. Due to the annual oscillation of
the intertropical convergence zone (ITCZ), dust transport also exhibits a seasonality in terms of
its geographic extent (Adams et al., 2012; Chin et al., 2014; Prospero & Lamb., 2003; Prospero, 1968;
Prospero & Mayol-Bracero, 2013; Yu et al., 2019; Zuidema et al., 2019). However, marine shallow
cumulus clouds form year-round in the tropical Atlantic regardless of dust transport. Thus, it is
important to focus on aerosol characteristics across a full seasonal cycle to obtain a thorough
understanding of the role aerosols play on cloud formation in the tropical Atlantic (McCoy et al.,

2022).

Few studies have attempted to fully understand which aerosols are the most prominent

CCN during both the boreal summer (when dust concentrations are at a maximum) and boreal
winter (when dust concentrations are at a minimum) in the tropical North Atlantic MBL. African
smoke is one particle type that may be important for CCN activation in the tropical North
Atlantic, yet has been understudied at dust receptor sites like Barbados (Wex et al., 2016). In
contrast to dust, previous research has shown that smoke particles are an important source of
CCN (Edwards et al., 2021; Lathem et al., 2013; Pierce et al., 2007; Spracklen et al., 2011) with
some research showing that smoke particles can activate at supersaturations as low as 0.05%
(Rogers et al., 1991).

There are a number of reasons that explain why smoke particles can be effective CCN.

Smoke particles are often complex mixtures of both organic and inorganic components that
change compositionally and morphologically during their residence time in the atmosphere
(Cappa et al., 2020; Hodshire et al., 2019; Konovalov et al., 2021; Reid et al., 2005; Wu et al.,
2021). Smoke properties may also vary between fires depending on fuel type and moisture,
combustion phase, wind conditions, etc. (Andreae, 2019; Miles et al., 1995; Reid et al., 2005). In
general, smoke particles are often found in the accumulation mode of the aerosol size
distribution and primarily contain particulate organic matter, black carbon, and inorganic
components including potassium chloride salts (Reid et al., 2005). Upon emission, smoke can
undergo chemical processing through photochemical and heterogeneous reactions, including the
loss of chloride and acquisition of sulfate and nitrate, creating potassium sulfate compounds in
smoke that are often used as tracers of aged smoke and can affect the hygroscopicity of smoke
particles (Capes et al., 2008; Hennigan et al., 2010, 2011; Reid et al., 2005; Zauscher et al.,
2013). Chemical processing can also lead to morphological changes as the condensation of
gaseous compounds and multiphase processes with aqueous compounds can result in the growth
and sphericity of smoke particles, which in turn can affect the CCN properties of smoke particles
(Abel et al., 2003; Giordano et al., 2015; Reid et al., 1998; Zhang et al., 2008). The variations in
the chemical and physical properties of emitted smoke particles as well as the changes these
properties can undergo in transit make it difficult to predict the ability of smoke particles to act
as CCN.

In this study, we investigated the relationship between submicron aerosol composition

and CCN in the tropical North Atlantic MBL during marine background conditions and
conditions affected by long-range continental aerosol transport of smoke and dust particles
(henceforth referred to as "CAT" conditions). To perform this work, we collected aerosol
samples and size-resolved CCN data from January to February 2020 at the Barbados

Atmospheric Chemistry Observatory (BACO) during the Elucidating the Role of Clouds–

Circulation Coupling in Climate/Atlantic Tradewind Ocean-Atmosphere Mesoscale Interaction

Campaign (EUREC$^4$A/ATOMIC) campaigns (Quinn et al., 2021; Stevens et al., 2021).

Conducting this research during the boreal winter provided a unique opportunity to explore

aerosol-cloud interactions in meteorological conditions different from those that are typically

studied in the tropical North Atlantic. Dust primarily arrives to Barbados during the summer

months with peaks in June and July (Zuidema et al., 2019). As a result, dust receptor sites in

Barbados have historically been used to compare CAT and marine background conditions during

the boreal summer. During the winter, the southward shift of the ITCZ directs African dust to

South America, resulting in a decrease in dust concentrations over Barbados during the winter

months with days in December and January sometimes receiving no dust at all (Prospero, 1968;

Prospero et al., 2014; Prospero & Lamb, 2003; Prospero & Mayol-Bracero, 2013). However, during the

EUREC$^4$A/ATOMIC campaigns, we observed anomalous wintertime transport of African

aerosols to Barbados, which provided novel sampling conditions to study the effects of various

aerosol types on cloud droplet formation. Specifically, we were able to explore how the addition

of continental aerosols like mineral dust and smoke particles to background marine aerosols

consisting of organics, sulfates, and sea salt affects CCN activity. This allowed us to compare the

impact of ocean-derived vs. long-range transported aerosol on water uptake properties and CCN

concentrations. We conclude this manuscript by discussing the importance of our findings for

cloud formation in the tropical North Atlantic.

**Methods**

Measurement Site and Sampling Period

Aerosol samples and size-resolved CCN data were collected at the Barbados
Atmospheric Chemistry Observatory (BACO) on Ragged Point during the EUREC[4]A and
ATOMIC field campaigns from January 20, 2020 -February 20, 2020 (Quinn et al., 2021;
Stevens et al., 2021). Ragged Point (13° 6' N, 59° 37' W), a prominence on Barbados' east coast,
is an ideal location for studying the impact of long-range African aerosol transport on aerosol-
cloud interactions as it is situated on the most easterly island in the Caribbean and is exposed to
the steady easterly trade winds. Thus, the east coast of the island is subject to little anthropogenic
aerosol influence from local islands to the west (Prospero et al., 2005; Savoie et al., 2002).
Further, the island is at a latitude coinciding with the outflow of African aerosols such as mineral
dust (Carlson & Prospero, 1972; Prospero, 1968) and smoke particles (Archibald et al., 2015) as well
as tropical marine cumulus clouds (Stevens et al., 2016).
Air Mass Origins
During the sampling period, air masses of varying compositions were observed at Ragged
Point. To determine the origin of these air masses, 150 h back trajectories were generated every 6
h at heights of 500, 1000, and 1500 m throughout the campaign using the NOAA Hybrid Single
Particle Lagrangian Integrated Trajectory (HYSPLIT) model calculated using model vertical
velocity and meteorology from the National Center for Environmental Prediction (NCEP) 1-
degree Global Data Assimilation System (GDAS) (Rolph et al., 2017; Stein et al., 2015).
Dust Concentration
To collect aerosols, BACO is equipped with a high-volume sampler and an isokinetic
aerosol inlet on top of a 17 m tall tower situated on a 30 m bluff along the coast at Ragged Point.
Daily dust mass concentrations were determined from filter-based measurements (Prospero et al.,
2021; Zuidema et al., 2019) using a high-volume air sampler pumping at a rate of approximately
0.7 $m^3$/min across a 20 cm x 25 cm cellulose Whatman-41 (W-41) filter with a nominal 20 μm
pore size. W-41 filters were chosen for this analysis as they allow high flow rates and yield a
collection efficiency of 95% or better for dust (Kitto & Anderson, 1988) and submicron aerosols
(Pszenny et al., 1993). Upper particle diameter limits for W-41 filters are approximately 80-100
μm or greater (Barkley et al., 2021). After aerosol collection, the filters are washed with milli-q
water three times to remove soluble material then placed in a furnace and combusted at 500°C
for about 12 h (i.e., overnight). Procedural blanks are also collected by placing a filter in the
sampler for 15 minutes without turning on the pump. The resulting ash mass from a sample
minus the mass of a filter blank is the gross ash weight, which is then adjusted by a factor of 1.3
to convert the ash weight to a mineral dust concentration. Previous research has confirmed the
validity of this method for determining dust mass concentrations through chemical analysis of
dust ash determined from filters collected in Barbados, crustal abundance, and soil dust
composition (Zuidema et al., 2019). A correction factor of 1.3 is applied to the calculated dust
concentrations to account for dust components such as bound water or soluble ions that are lost
during the heating process (Prospero, 1999; Zuidema et al., 2019).
Aerosol Chemical Composition
Aerosol particles were sampled at ambient relative humidity (RH) through an isokinetic
aerosol inlet and collected using a three-stage microanalysis particle sampler (MPS-3, California
Measurements, Inc.), which samples particles from diameters of 5.0-2.5 μm (stage 1), 2.5 μm –
0.7 μm (stage 2), and <0.7 μm (stage 3). For each set of samples (1 set including 1 sample from
each stage of the MPS), the MPS was run for 45 min at 2 L/min flow starting at approximately
9:30 local time or 13:30 coordinated universal time (UTC). Meteorological data from a local
station was also used to manually check that wind direction fell between 335° and 130° and wind
speeds were greater than 1 m/s during all sampling periods. Sampling during these wind
conditions ensures that only air from the open ocean was sampled rather than local,
anthropogenically-influenced air.

To determine aerosol chemical composition, particles were deposited onto carbon-

coated copper grids (Ted Pella, Inc.) that were later analyzed at the Pacific Northwest National
Laboratory using computer-controlled scanning electron microscopy coupled with energy
dispersive x-ray spectroscopy (CCSEM/EDX; Quanta 3D) to determine the elemental
composition of individual particles. We also collected samples on silicon wafers (Ted Pella, Inc.)
which were analyzed with CCSEM/EDX to confirm the vailidity of the carbon measurements on
the carbon-coated copper grids. Here, we focus only on the submicron particle population which
exerts a greater influence on CCN number concentrations and is more sensitive to chemical
changes that affect its hygroscopicity. Thus, for this study we focus primarily on data from stage
3 of the MPS, representing <0.7 μm diameter particles.

CCSEM/EDX is a valid method for determining size-resolved chemistry of the aerosol

loading as CCSEM excels in calculating particle size by imaging individual aerosols while EDX
provides the relative abundances for elements of interest (Tomlin et al., 2021). Percent
composition threshold values of 1% were used to ensure the presence of elements detected by the
EDX. Single-particle analysis using CCSEM/EDX was limited to 16 elements found in common
aerosols such as dust, sea salt, and smoke particles: carbon (C), nitrogen (N), oxygen (O),
sodium (Na), magnesium (Mg), aluminum (Al), silicon (Si), phosphorus (P), sulfur (S), chlorine
(Cl), potassium (K), calcium (Ca), vanadium (V), manganese (Mn), iron (Fe), and nickel (Ni).
The EDX peak for Cu is heavily influenced by the background signal from the Cu grid and is
excluded from analysis. Samples collected on Si substrates confirmed the validity of the C signal
in analyzed particles, as the carbon coating on the Cu substrates has the potential to generate a
background signal as well. An excess of 1000 particles were analyzed in each sample. Due to
size limitations of the CCSEM, only particles with diameters >0.1 μm were analyzed. Data
products from CCSEM/EDX analysis were then analyzed in MATLAB (ver 9.6.0; The
Mathworks, Inc.) using a K-means clustering algorithm (Ault et al., 2012; Shen et al., 2016). The
algorithm operates by generating categories of similar particles (clusters) based on the presence
and intensity of elemental peaks in individual single-particle EDX spectra. These clusters are
then assigned to particle types based on their size, morphology, characteristic EDX spectra, and
existing literature. A more thorough explanation of the k-means clustering algorithm and particle
identification process including the plots used to perform particle identification (Figure S1) is
provided in the Supporting Information (SI).
Size-Resolved CCN Measurements and Data Analysis
To determine the size-resolved CCN activity of aerosol particles during the sampling
period, we used a continuous-flow streamwise thermal gradient CCN counter (CCNC, model
CCN-100, DMT, Longmont, Co, USA; (Roberts & Nenes, 2005; Rose et al., 2008)) combined with
a differential mobility analyzer (DMA, modified model M, Grimm Aerosol Technik, Ainring,
Germany) and condensation particle counter (CPC, model 5412, Grimm Aerosol Technik). The
method is described in detail in Pohlker et al. (2016). Flows for the size-resolved CCN set-up
included a sheath:sample flow ratio of 10 for the CCN counter (sample flow rate of 0.5 L/min), a
sheath flow of 8 L/min for the DMA, and a sample flow of 0.6 L/min for the CPC. Upon entering
the system, the sampled air was dried using a condensation drier to maintain a relative humidity
(RH) between 20 and 30% and to ensure reliable hygroscopicity measurements. After drying, the
particles passed through a DMA which selected particles with a diameter (D) between 20 and
245 nm. The monodisperse aerosol-laden flow was then split between the CCNC and CPC.
Inside the CCNC, the particles were subjected to supersaturations (S) of 0.09, 0.16, 0.24, 0.43,
and 0.74 %.
Calibrations of the CCNC supersaturations were performed according to the method
described in Rose et al. (2008) by generating and size-selecting ammonium sulfate particles that
were analyzed by the CCNC set to a designated temperature gradient as well as a CPC to
measure total condensation nuclei (CN) values. Plots comparing CCN/CN and dry particle
diameter were then used to determine the diameter at which 50% of the particles in an aerosol
population activate as CCN at a particular S, also called the critical activation diameter ($d_{50}$). $D_{50}$
values were then used to determine supersaturation. Supersaturations were plotted against the
designated temperature at the calculated supersaturation. The resulting plot provided a linear
curve that could be used to adjust the supersaturation shown by the instrument to the actual value
of the column supersaturation. After calibrating, S values averaged 0.08, 0.15, 0.23, 0.41, and

247 0.71%.

For ambient sampling, particles that activate as CCN at each S and D are counted in the
CCNC as CCN, while all particles of a selected D are counted in the CPC to determine the total
aerosol concentration of particles at each D. By scanning D at a given value of S, measurements
from the CPC and CCNC are then used to generate an activation curve used to calculate the $d_{50}$.
These values, along with the particle number size distribution determined by an SMPS (SMPS,
Grimm model 5420 with CPC 3772) operating independently of the CCNC set-up, are then used
to calculate the effective hygroscopicity parameter κ using equation (1) according to the κ-
Köhler model (Petters & Kreidenweis, 2007):
$$\kappa = \frac{4A^3}{27D_p^3 \ln^2 S_{crit}} \qquad (1)$$

where $D_p$ is the dry particle diameter, $S_{crit}$ is the supersaturation set by the CCN counter, and A is
the Kelvin term calculated from equation (2):
$$A = \frac{4\sigma M_w}{RT\rho_w} \qquad (2)$$

Where $\sigma$ is the surface tension ($\sigma=0.072$ J/m$^2$), R is the universal gas constant, $M_w$ is the
molecular weight of water, and $\rho_w$ is the density of water. In the $\kappa$-Köhler model, higher values
of $\kappa$ indicate a more hygroscopic particle that is more efficient at taking up water and can
activate as CCN at lower S.

**Results and Discussion**

In this section we will show that, upon arrival of co-transported dust and smoke, smoke
originating from fires in the African Sahel dominate the accumulation mode particle population
in the tropical North Atlantic MBL, which results in an increase in CCN number concentration.
Though dust and smoke are both transported to the region, smoke dominates the accumulation
mode number concentration by an order of magnitude compared to dust. These findings are
supported by data products from dust mass concentrations, size-resolved hygroscopicity, single
particle data (CCSEM-EDX), and air mass history (NOAA'S HYSPLIT model), which all
complement one another and provide unique insights into the aerosol sources, their single
particle composition, and their effects on cloud droplet activation.
Air Mass Characteristics during the EUREC$^4$A and ATOMIC Campaigns
To confirm the origins of the various air masses sampled, we performed back trajectory
analysis throughout the campaign using NOAA's HYSPLIT model (Figure 1) and quantified
dust mass concentrations (Figure 2a). Results of these two analyses show that Barbados was
influenced by two types of air masses during the sampling period: air masses that, over the
course of 6 days, do not pass over land (referred to as clean marine conditions), and air masses
that have passed over the African continent (referred to as continental aerosol transport (CAT)
events). Back trajectory analysis was not conducted for time periods longer than 6 days, which
introduces the possibility that marine air masses could have been influenced by European
outflow as well. Figure 1 shows that during periods with low dust mass concentrations and a
bimodal size distribution, air masses originated from the remote Atlantic Ocean at higher
latitudes with no land contact over 6 days. During time periods with high dust mass
concentrations, air masses originated from continental Africa. Figure 2a shows that the total
mass concentration of dust particles correlates very well with the arrival of air masses originating
from Africa. During time periods when dust concentrations were low, the particle loading has a
bimodal size distribution characteristic of clean marine air masses (Figure 2b; Ault et al., 2013;
Hoppel et al., 1986; O'Dowd et al., 2004). Upon the increase in dust mass concentrations, the
submicron particle size distribution becomes unimodal and the smallest Aitken size mode is
negligible, suggesting that long-range transported (LRT) particles are either dominant over the
background marine particle loading or that smaller Aitken mode particles are coagulating onto
larger LRT continental aerosols to form a unimodal accumulation mode (Tomlin et al., 2021).

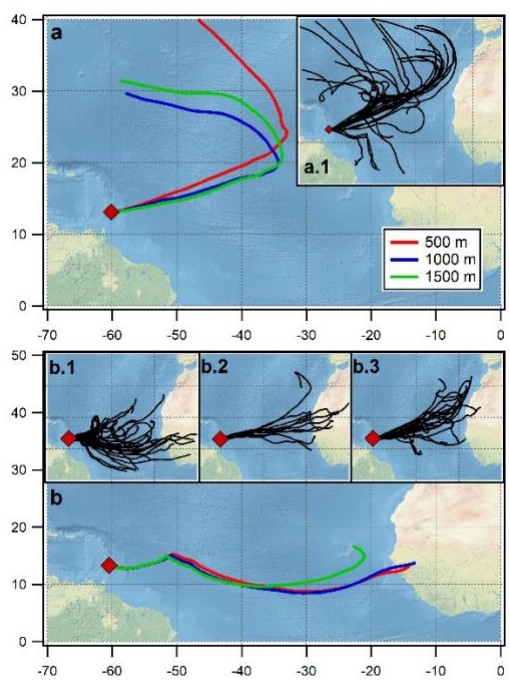


**Figure 1**: HYSPLIT back trajectories at Ragged Point, Barbados (red diamond) for the
EUREC[4]A/ATOMIC field campaign. (a) Back trajectories for 2020/2/8 18:00 UTC at heights of
500 m (red), 1000 m (blue), and 1500 m (green) exemplify air mass origins during clean marine
sampling conditions. Subplot a.1 shows all back trajectories from clean marine sampling
conditions collected at 6 h intervals with a release altitude of 1000 m from 2020/1/29 0:00 –
2020/1/29 12:00, 2020/2/6 12:00 – 2020/2/9 18:00 and 2020/2/12 12:00 – 2020/2/15 6:00 UTC.
(b) Back trajectories for 2020/2/2 18:00 UTC at 500 m, 1000 m, and 1500 m exemplify air mass
origins during CAT conditions. The subplots, b.1, b.2, and b.3 show all back trajectories for 3
time periods during which continental aerosols were sampled: b.1) 2020/1/29 18:00 – 2020/2/6
6:00, b.2) 2020/2/10 0:00 – 2020/2/12 6:00, and b.3) 2020/2/15 12:00 – 2020/2/20 18:00 UTC.
Trajectories for b subplots were also collected at 6 h intervals with a release altitude of 1000 m.

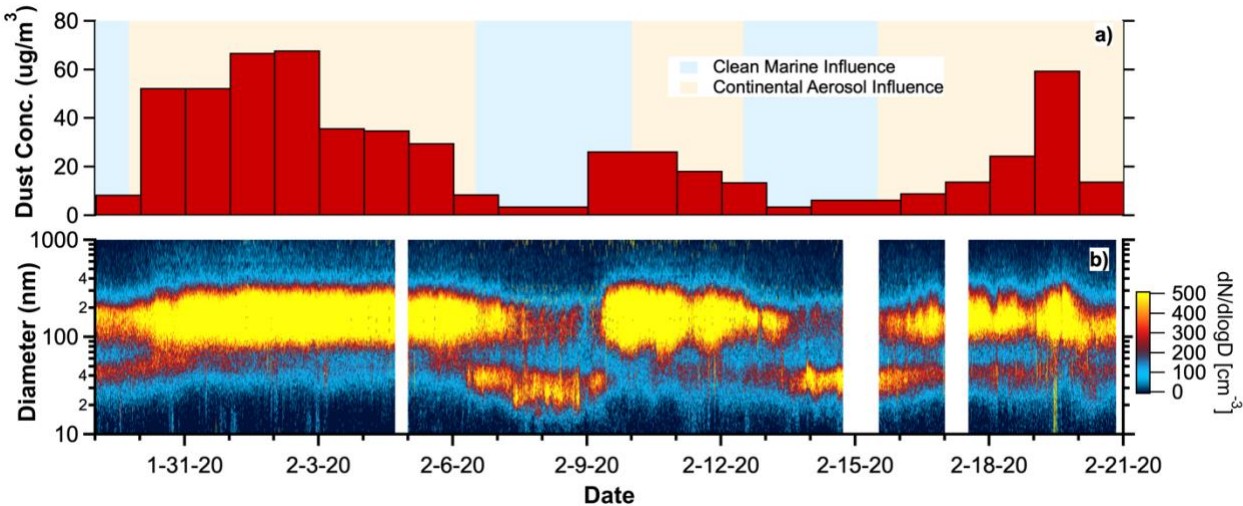


**Figure 2** – Temporal evolution of (a) dust mass concentrations determined from bulk aerosol
filter samples and (b) submicron aerosol particle size distributions determined with an SMPS.
Time for both plots is given in UTC (-4 h local Atlantic Standard Time). Color shading in (a)
represents continental aerosol influence (orange shading) and clean marine influence (b) as
determined by NOAA HYSPLIT back trajectories calculated at Ragged Point.

Single Particle Aerosol Composition
CCSEM/EDX analysis from the EUREC[4]A and ATOMIC campaigns revealed the presence
of several particle types with distinct chemistries and morphologies in the submicron aerosol
loading (Ault et al., 2014; Behnke et al., 1997; Gaston et al., 2011a, 2013a). Figure 3 presents
SEM images (left) and EDX spectra (right) for each particle type detected on stage 3 of the MPS
(particle diameter <0.7 µm), including sea spray, aged sea spray, mineral dust, internally mixed
mineral dust and sea spray, sulfate, smoke, internally mixed mineral dust and smoke, and
organics. Sea spray particles as well as internally mixed mineral dust and sea spray particles
were dominant components of the supermicron aerosol loading but are only minor components
of submicron aerosol.

*Sea Spray*

Sea spray particles were characterized by high relative abundance of approximately equal

parts Na and Cl, indicating the formation of halite (NaCl). Morphologically, sea spray particles
have a cubic shape that represents the crystal structure of halite. Small Mg peaks approximately
10% of the height of Na peaks were also observed in NaCl particles and reflect the Na:Mg ratio
of seawater. Additional components of sea spray particles include rod-shaped particles
containing Ca and S (presumably calcium sulfate) that were often found attached to NaCl
particles (Ault et al., 2013; Bondy et al., 2018; Choël et al., 2007). Elements such as N and S that
may suggest aging of sea spray were either absent or present in small relative abundance on
NaCl components of sea spray particles.

*Aged Sea Spray*

Aged sea spray was defined by the presence of sea salt components including Na, Mg, K, S,

and Cl. In contrast to freshly emitted sea spray particles, aged sea spray has a characteristically
low or absent Cl signal with a strong presence of N or S. Figure 3 provides an example of an
aged sea spray particle in which Na is high (indicating the presence of salt), but with a low Cl
peak (suggesting the particle has been aged). The presence of S in this spectrum may explain the
low relative abundance of Cl compared to Na. Sea spray can be aged through reactions with
sulfuric acid ($H_2SO_4$), dinitrogen pentoxide ($N_2O_5$), and/or nitric acid ($HNO_3$) which results in Cl
depletion and S or N enrichment (Ault et al., 2014; Ault, Guasco, et al., 2013; Behnke et al.,
1997; Gaston et al., 2011, 2013; Sobanska et al., 2003). Morphologically, aged sea salt particles

had either a similar appearance to fresh sea salt particles, which is often either cubic (as seen in

Fig 3), or appeared as a flakey amorphous mass (Hoffman et al., 2004; Laskin et al., 2012; Li et

al., 2010).

*Mineral Dust*

Mineral dust is characterized by the presence of aluminosilicate elements such as Si, Al, Fe,

K, Ca, and Mg in EDX spectra, which is consistent with previous studies of African dust

(Denjean et al., 2015; Hand et al., 2010; Krueger et al., 2004; Levin et al., 2005; Twohy et al.,

2009). Elements such as S and N were not observed in this particle type (Kandler et al., 2018)

suggesting that detected dust had not undergone chemical processing during transport. Dust often

appeared as a flakey or nodular amorphous mass as exhibited in Fig 3 and previous literature

(Krueger et al., 2004; Laskin et al., 2005; Pachauri et al., 2013; Remoundaki et al., 2011).

*Internally Mixed Mineral Dust and Sea Spray*

Particles containing elements indicative of both mineral dust (Si, Al, Fe, K, Ca, and Mg) and

sea salt (approximately equal relative abundances of Na and Cl) were characterized as internally

mixed mineral dust and sea spray (Choël et al., 2007; Deboudt et al., 2010; Sobanska et al.,

2014). Elements such as S and N were often not present in this particle type, suggesting the

particles had not undergone atmospheric aging during transit. Particles containing both dust and

sea spray components often appeared as conglomerates of multiple particles with some parts

containing more sea spray components and others containing more mineral dust components.

*Sulfate*

Sulfate-rich particles are a prevalent component of marine submicron aerosol (O'Dowd & de

Leeuw, 2007) and characterized here by a dominant S component - often with high relative

abundance of C, O, and N. These particles are likely sulfates bound to $NH_4^+$ such as ammonium
sulfate $((NH_4)_2SO_4)$ or ammonium bisulfate $(NH_4HSO_4)$ (Hand et al., 2010). The high relative
abundance of C indicates a large organic fraction as well. The morphology of sulfate particles
appeared smooth and spherical as reported in previous literature (Nájera & Horn, 2009).
*Smoke*
Smoke particles were identified by the presence of C with K and S likely representing
internally mixed organic and black carbon with potassium-containing salts. K is a well-known
indicator for biomass burning (Andreae, 1983; Hand et al., 2010; Hudson et al., 2004; J. Li et al.,
2003; Murphy et al., 2006; Pósfai et al., 2003), especially in flaming conditions in Savannah fires
as opposed to smoldering conditions (Echalar et al., 1995; Maenhaut et al., 1996).
Morphologically, smoke particles can be spherical due to aging or coatings but can also appear
as aggregates or chains of spheroids (Dang et al., 2021; Hand et al., 2010; Miller et al., 2021;
Pósfai et al., 2003). In this study, smoke particles most frequently appeared as small spherical
particles.
*Internally Mixed Mineral Dust and Smoke*
Internally mixed mineral dust and smoke particles are characterized by dust components such
as Si, Al, Fe, Ca, and Mg with strong contributions of K, S, C, and O. Morphologically, internal
mixtures of dust and smoke appear as aggregates of amorphous dust particles with clusters or
spheres representing soot from smoke. Single particle chemical analysis of these particles show
distinctions between the dust and smoke portion of the particle, with the dust portion having
typical dust components (Si, Al, Fe, Ca, and Mg) and the smoke portion having typical smoke
components (K and S with C and O). Previous research has observed internal mixing of
carbonaceous particles and dust particles in Africa when significant amounts of both biomass
burning and dust were present (Hand et al., 2010); however, we show that these internal mixtures
can be transported all the way to the Caribbean as well.
*Organics*
Organic particles are defined by strong signals of C and O with few other elements present, if
any (Hand et al., 2010). The absence of S or N, which are often indicative of sulfate and nitrate,
respectively, suggests that these particles have undergone minimal chemical aging.
Morphologically, organic particles are characterized as small individual spheres. The organics
were likely marine in origin (Russell et al., 2010) as they were the smallest particle type
observed both during clean marine conditions and during CAT conditions, which indicates they
may be a "background" aerosol type (Russell et al., 2010).

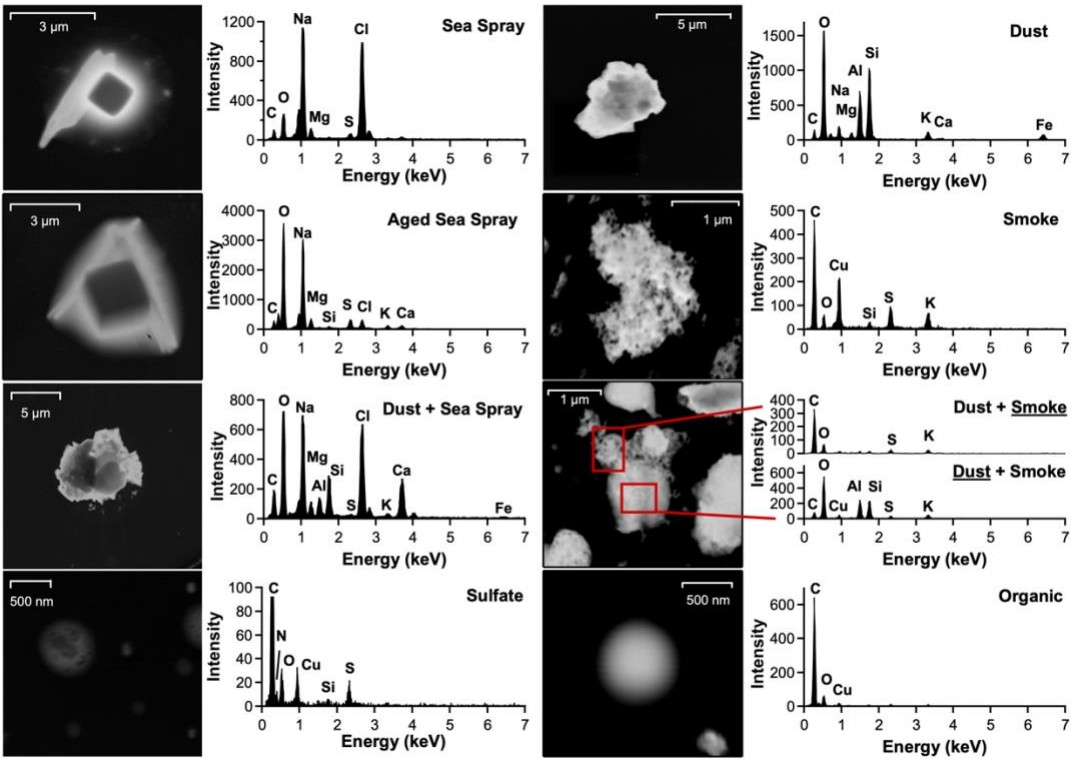


**Figure 3**: Characteristic aerosol particle types observed by SEM-EDX images (left) and spectra (right) in samples collected during the EUREC$^4$A/ATOMIC campaign. Spectra for the Dust + Smoke particle type represent different areas analyzed on the particle with EDX, denoted by the red boxes.

Arrival of Anomalous Wintertime Co-Transported Dust and Smoke

Figure 4 presents number fractions for particles detected in the submicron aerosol throughout the sampling period and reveals a similar trend in smoke particle number fractions to those of dust mass concentrations in Fig. 2, suggesting that smoke and dust were co-transported to Barbados from Africa. A similar plot to Fig. 4 that contains temporal chemistry from stage 1 and stage 2 of the MPS (representing supermicron particles >0.7 μm diameter) determined using CCSEM/EDX analysis can be found in the SI (Figure S2). During the boreal winter, the Sahel

region in North Africa experiences its fire season in which large swathes of land are burned and
large plumes of smoke are emitted from the region (Figure S3; Ansmann et al., 2009; Barkley et
al., 2019; Roberts et al., 2009). However, due to the southward shift in the ITCZ during boreal
winter, smoke is expected to be transported primarily to South America (Moran-Zuloaga et al.,
2018; Talbot et al., 1990; Wang et al., 2016). In our study, we observed the arrival of this smoke
on Barbados. These findings are supported by temporal carbon monoxide (CO) column density
measurements that are often used as a tracer for smoke (Figure S4 and S5). Periods that
correspond to clean marine influence are dominated by sulfate and organic particles in the
submicron aerosol (Figure 4). Upon arrival of continental aerosols, wildfire smoke appeared to
dominate the number fraction of submicron aerosol.

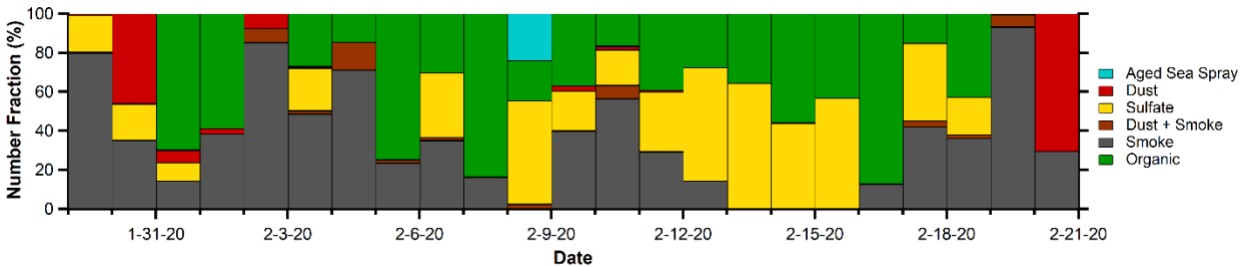

**Figure 4**: Temporal evolution of submicron number fractions for different types of aerosol particles determined by CCSEM/EDX analysis. The total number of particles analyzed for each day ranges from 1000 to 20,000.

Figure 5 presents size-resolved chemical data from CCSEM/EDX analysis from clean
marine periods (average of all clean marine periods) and one exemplary time period influenced
by continental air masses (CAT event 1). Similar plots for other CAT events are provided in the
SI (Figure S6). Average particle diameters for each particle type detected during each sampling
period are also provided in Table S1. Figure 5 shows that in clean marine periods, a small
fraction of large particles have both a smoke and a dust signature. This suggests that our "clean
marine conditions" are only "clean" relative to time periods dominated by dust and smoke, rather
than pristine clean marine conditions without any continental aerosol influence. The CAT event
plot in Fig. 5 demonstrates that dust as well as internally mixed dust and smoke particles
dominate in the submicron aerosol loading, followed by smoke. Smoke particles follow as the
next largest particle type. Organics dominate in the smallest size fractions, followed by sulfates,
suggesting a secondary source for sulfate (Bates et al., 1992). Aged sea salt particles were on
average smaller than most dust, internally mixed dust and smoke, and smoke particles. Figure 5
also shows that at a diameter of ~0.1µm (which is approximately the $d_{50}$ of CCN at S 0.16% in
clean marine conditions and CAT conditions) the composition is dominated by sulfates and
organics in the clean marine conditions, while smoke and organics dominate in the CAT event.
The large decrease in sulfate number fraction during CAT events might be caused by the
condensation of Aitken mode sulfate-containing particles onto larger, long-range transported
particles as indicated in Fig. 2 (Gaston et al., 2010).

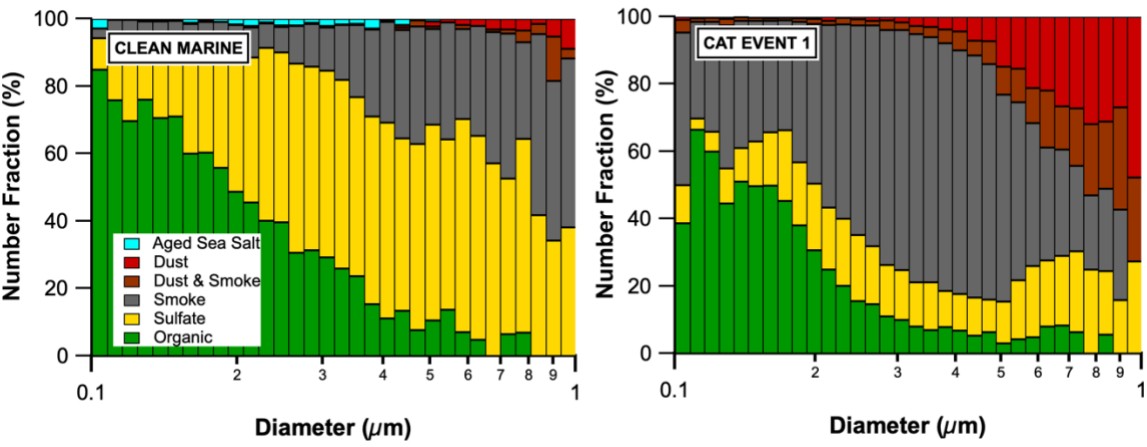


**Figure 5:** Number fractions of different types of submicron aerosol particles plotted against
particle diameter. The "clean marine" plot (left) includes data from all clean marine sampling
periods. The "CAT Event 1" plot (right) includes data from the first period in which dust and
wildfire smoke were observed over Barbados (2020/1/29 18:00 – 2020/2/6 6:00 UTC). Particles
were organized into 32 size cuts (bins) to maximize resolution of size-resolved chemical data.
Particle counts in each bin range from 34 particles to up to 3041 with an average bin count of
493 particles for the Clean Marine plot and 973 for the CAT Event plot.

Changes in Aerosol Hygroscopicity during EUREC[4]A/ATOMIC

Comparisons between $\kappa$ and submicron single particle elemental composition reveal that

smoke particles lower the hygroscopicity of the submicron aerosol  compared to marine-derived
submicron aerosol in the tropical North Atlantic. Figure 6 presents boxplots for $\kappa$ values as well
as average $d_{50}$ measured at each S during both clean marine conditions and CAT conditions.
Both plots show a similar trend in which average $\kappa$ increases from 0.09% S to 0.24% S. Then,
with each subsequent increase in S after 0.24% S, $\kappa$ decreases as smaller, less hygroscopic
particles activate at higher supersaturations. The low hygroscopicity of these smaller particles
can be explained by compositional changes in the aerosol loading exhibited in Fig 5, reflecting
the shift in particle chemistry from mostly sulfate to mostly organic with decreasing particle size.
Also of note is the $\kappa$ of 0.6 observed for clean marine conditions at 0.24% S, which matches $\kappa$
measurements for ammonium sulfate particles that can dominate along with sea spray organics
during clean marine conditions (Petters & Kreidenweis, 2007).

There is also a noticeable drop in average $\kappa$ between the same supersaturations in clean

marine conditions compared to CAT conditions. For example, at 0.16% S, $\kappa=0.52\pm0.09$ for all
clean marine condition periods and $\kappa=0.46\pm0.10$ for all continental aerosol transport periods.
This is likely due to the addition of less hygroscopic material such as dust and smoke particles
that are not dominant in clean marine conditions. As expected, trends in average $d_{50}$ for both
plots indicate that smaller particles activate as CCN with larger supersaturations. Activation
diameters during CAT conditions are also larger than corresponding activation diameters in clean
marine conditions for the same supersaturation. This also suggests that the addition of less
soluble material from transported smoke particles lowers the hygroscopicity and increases the
activation diameter.

When comparing hygroscopicity data from this study to previous research, we find both

similarities and differences in κ trends. For example, Good et al. (2010) presents data collected in
the tropical eastern Atlantic that provides an ideal comparison to our findings. On average, their
values for κ in clean marine conditions and during observations of dust transport (κ=1.15-1.4 and
0.8-0.92, respectively) were much higher than our observed values. However, Good et al. (2010)
shows similarities to our work through the distinct drop in κ between clean marine conditions
and CAT conditions which is attributed to the addition of hydrophobic dust to the aerosol
loading. Wex et al. (2016) present CCN data from ground-based field sampling in November and
April at Ragged Point, Barbados. They show a similar trend in κ in which values increase from
0.1% S, peak at 0.2% S, then decrease with each subsequent increase in S. They also found a
similar drop in κ upon the arrival of long-range transported aerosols, likely due to less
hygroscopic particles from continental sources activating as CCN. A separate study from
Kristensen et al. (2016) conducted similar research at Ragged Point, Barbados during the boreal
summer. The range in κ values of 0.2-0.5 match those observed in our work, especially during
CAT events. However, Kristensen et al. (2016) determined that concentrations of dust, sea salt,
and soot were too small to influence CCN, concluding that sulfates and organics were the
primary CCN types. They conclude that the low κ values observed during their sampling were
due to organic compounds activating as CCN. We find similar κ values to Kristensen et al.
(2016) and a similar particle chemistry of the accumulation and Aikten modes during clean
marine conditions, suggesting that organics and sulfates were the primary CCN types during
clean marine conditions studied for the EUREC[4]A and ATOMIC campaigns as well. We also
observe a drop κ from clean marine conditions to CAT events. This drop in κ indicates the
influence of an additional CCN particle type contributed by the CAT events.

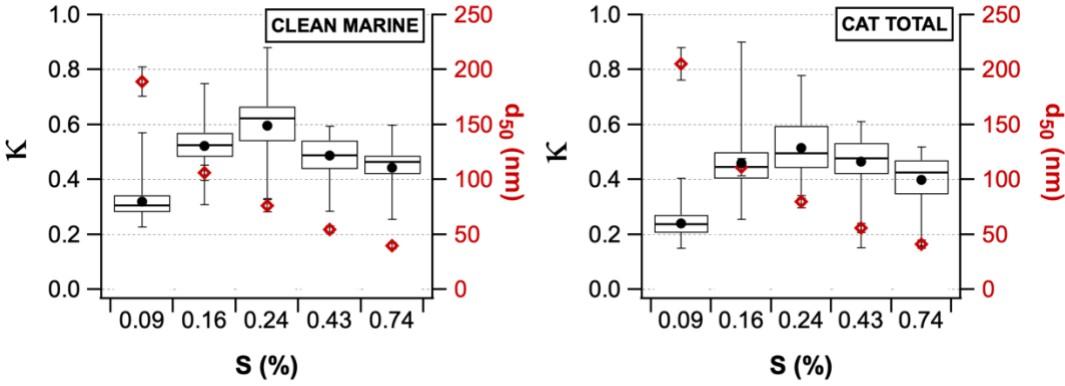


**Figure 6:** Hygroscopicity parameter κ (left axis, box plots) and corresponding mean
diameter at which 50% of the particles in an aerosol population activate as CCN at a particular S,
also called the critical diameter "$d_{50}$" (right axis; red markers) for the investigated
supersaturations (S). Whiskers on "$d_{50}$" markers represent standard deviation values of "$d_{50}$".
Black dots in the boxplot indicate κ mean values. Boxes represent the upper quartile, median,
and lower quartile κ values at each S. Whiskers represent the upper and lower limit of κ at each
S.


African Smoke Particles Enhance CCN Concentrations
Comparisons between smoke fractions and CCN counts suggest that smoke particles
enhance the number of CCN in the tropical North Atlantic MBL. Figure 7 presents two temporal
plots of κ (Figure 7a) and smoke number fractions with CCN counts measured at 0.16% S
(Figure 7b). Table 1 provides averages of CCN concentrations for each time period shown in
Fig. 7b. Table S2 also provides average and median counts of analyzed smoke particles
calculated for each CAT event, including CAT Event 3 which had fewer smoke particles and a
higher κ compared to other CAT Events. Figure 7 suggests that there is an inverse relationship
between κ and smoke number fractions in which an increase in smoke particles results in a
decrease in κ. This is likely due to the activation of smoke particles as CCN, which are on
average less hygroscopic than the sulfate particles that act as CCN during clean marine
conditions. Figure 7b shows a positive correlation between smoke number fractions and CCN
counts. A correlation plot of smoke number fraction and CCN concentrations is also provided in
Fig. S7 to further emphasize their direct relationship.
There are multiple possible explanations for why African smoke particles may have acted
as CCN. As shown in Fig. 5, smoke particles are larger than organics and sulfates, on average,
and dominate sulfate and organic particle number concentrations upon arrival of long-range
transported African aerosols. In this case, the relatively large size of the smoke particles makes
for better CCN compared to organics or sulfates via the Kelvin effect (Dusek et al., 2006). The
presence of salts in smoke particles has also been shown to be an important component in smoke
hygroscopicity and may explain why smoke is efficient as CCN. Previous studies have shown
that smoke particles often contain hygroscopic salts such as potassium chloride, potassium
sulfate, and potassium nitrate (e.g., KCl, KNO$_3$, and K$_2$SO$_4$) (Dang et al., 2022; Freney et al.,
2009; Zauscher et al., 2013). Other research also shows that only small fractions of salts are
needed to increase aerosol hygroscopicity (Roberts et al., 2002). Based on CCSEM/EDX
analysis, we find evidence that potassium sulfate salts may be present in transported smoke
particles, and thus may explain the potential for smoke to act as CCN.

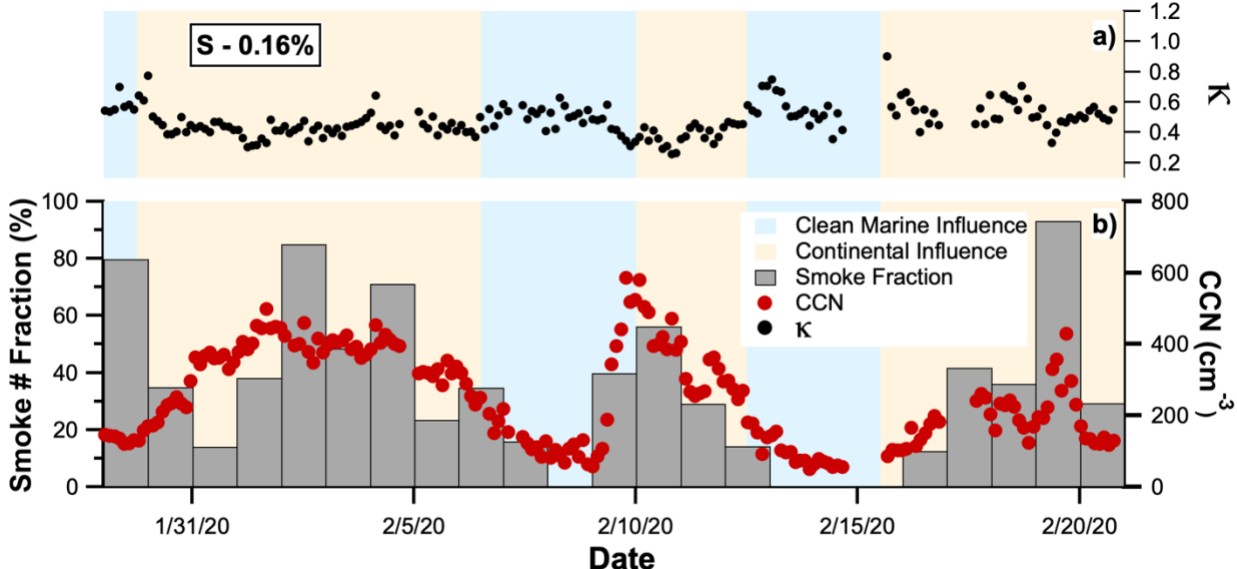


**Figure 7**: Temporal evolution of hygroscopicity parameter κ (black dots, upper panel) and CCN
number concentration (red dots, lower panel), both measured at S = 0.16%, and smoke particle
number fraction (grey bars, left axis, lower panel). Background color shadings indicate periods
of continental influence (orange) and clean marine influence (blue) determined by HYSPLIT
back trajectories and dust mass concentrations.
**Table 1** – Values for average CCN Concentrations and κ measured at 0.16% S during each clean
marine influence period and CAT event sampled during the EUREC[4]A and ATOMIC
campaigns.

| Sampling Period | Day/Time | CCN Concentrations (pt/cm$^3$) | Average κ |
|---|---|---|---|
| Clean Marine Period 1 | 2020/1/29 0:00 – 2020/1/29 12:00 | 140±10 | 0.58±0.07 |
| CAT Event 1 | 2020/1/29 18:00 – 2020/2/6 6:00 | 340±90 | 0.44±0.08 |
| Clean Marine Period 2 | 2020/2/6 12:00 – 2020/2/9 18:00 | 150±98 | 0.50±0.10 |

| | | | |
|---|---|---|---|
| CAT Event 2 | 2020/2/10 0:00 – 2020/2/12 6:00 | 400±106 | 0.38±0.06 |
| Clean Marine Period 3 | 2020/2/12 12:00 – 2020/2/15 6:00 | 100±42 | 0.55±0.10 |
| CAT Event 3 | 2020/2/15 12:00 – 2020/2/20 – 18:00 | 190±75 | 0.54±0.10 |


**Conclusions**

During clean marine conditions, the submicron aerosol loading consists primarily of
sulfate and organic particles. CCN measurements determine cloud activation by particles
approximately 80 nm in activation diameter with an average $\kappa = 0.52\pm0.08$ for 0.16% S.
Comparisons between particle size, hygroscopicity, and single particle elemental composition
suggest that sulfate particles (likely ammonium sulfate) are the primary CCN particles in clean
marine conditions. During the EUREC[4]A/ATOMIC campaign, Barbados received three African
aerosol transport events during which we detected mineral dust and smoke particles from
northern Africa. Upon the arrival of African aerosols to BACO, CCN average activation
diameter increased to approximately 200 nm while the average hygroscopicity of activated
particles for all CAT events decreased to $\kappa = 0.45\pm0.1$ for 0.16% S. Upon arrival of high
concentrations of smoke particles to Barbados, smoke particles dominate the accumulation mode
particle loading, decrease aerosol hygroscopicity, and also increase CCN number concentrations,
which could also increase the cloud droplet number concentration and alter cloud radiative
properties (Twomey, 1974). Overall, we find that smoke has a large effect on CCN number
concentrations during the boreal winter when smoke transport is high.

The observation of smoke transported to Barbados during the boreal winter also indicates
the large geographic extent of African smoke that can impact the MBL. Building upon recent
work from Ragged Point and other parts of the tropical and subtropical Atlantic (Holanda et al.,
2020; Kacarab et al., 2020; Schill et al., 2020; Zuidema et al., 2018) this work also indicates a

need for greater consideration of the impacts of smoke in the MBL, especially during the boreal
winter. Previous research conducted at Ragged Point has primarily focused on African dust,
which reaches its maximum during the boreal summer when smoke transport is low (Zuidema et
al., 2019). To better contextualize our findings, we analysed carbon monoxide column density (a
tracer for smoke) as well as aerosol optical depth (AOD; a tracer for dust and smoke) from 2018-
2022 (Figure S4 and S5). Figure S4 shows the temporal trends while Fig. S5 show seasonal
averages. As expected, AOD peaks in July when dust transport reaches a maximum. However,
Fig. S4 and S5 indicate that smoke is decoupled from dust, reaching a maximum in the spring
around April and a minimum in the summer when dust transport is highest. This finding suggests
that while the dust transport during the EUREC4A/ATOMIC campaigns is higher than average
dust loadings during this month (Zuidema et al., 2019), the amount of smoke observed is not
unique, but rather characteristic of the region. This is consistent with observations in Amazonia,
where smoke and dust transport during boreal winter and spring has been found consistently
since the first measurement campaign in 1987 (Talbot et al., 1990; Andreae et al., 2015; Moran-
Zuloaga et al., 2018) . Further, wintertime aerosol transport is typically transported at lower
altitudes as the height of emission for wintertime aerosols is lower compared to summertime
aerosol transport, leading to greater mixing into the MBL (Gutleben et al., 2022; Tsamalis et al.,
2013). Thus, smoke may be playing an important role on CCN formation throughout a large
portion of the year. This is especially true considering the large size of long-range transported
smoke plumes that have a wide geographic extent in which they can affect cloud formation. To
conclude, this work highlights the need to characterize African smoke transport to Ragged Point
and better understand the role of smoke in cloud formation, radiative forcing, and climate
(Pechony & Shindell, 2010; Shindell et al., 2009).

**Data Availability**

The data will be made publically available in the University of Miami data repository and will be linked with a doi.

**Author Contributions**

Conceptualization of this work was done by HMR, MLP, OK, and CJG. Collection of samples was conducted by HMR, OK, EB, and PS, while analysis was done by HMR, MLP, OK, NNL, and ZC. The development of the methods used in this work was done by HMR, MLP, OK, ZC, SC, APA, and CJG. Instrumentation used to conduct this work were provided by MLP, SC, APA, and CJG. Formal analysis of data was performed by HMR, MLP, CP, and OK. Validation of data products was performed by HMR, ZC, SC, APA, and CJG. Computer code used for data analysis was provided by MLP, OK, and APA. Data visualization was performed by HMR, MLP, and OK. PKQ, PZ, CP, and UP helped interpret results. Supervision and project administration duties were done by MLP and CJG. HMR wrote the original draft for publication, and all co-authors reviewed and edited this work.

**Competing Interests**

Some authors are members of the editorial board of Atmospheric Chemistry and Physics. The peer-review process was guided by an independent editor, and the authors have no other competing interests to declare

**Acknowledgements**

 C.J.G. acknowledges an NSF CAREER award (1944958). A portion of this research was performed on project awards (10.46936/lser.proj.2019.50816/60000110 and 10.46936/lser.proj.2021.51900/60000361) from the Environmental Molecular Sciences Laboratory, a DOE Office of Science User Facility sponsored by the Biological and Environmental Research program under Contract No. DE-AC05-76RL01830. P.K.Q. acknowledges PMEL contribution number 5353. MLP and CP acknowledge support by the Max Planck Society. P.Z. acknowledges support from the NOAA grant OAR CPO NA19OAR4310379.

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
