# Peer review of "African Smoke Particles Act as Cloud Condensation Nuclei in the Wintertime Tropical"

_Atmospheric Chemistry and Physics, 2022_

## Author Comment (AC1)

**We thank the reviewers for their helpful suggestions and comments. Our responses to reviewer comments are black normal text indented 0.5" and revised text as it appears in the manuscript is in blue. Line marks in the manuscript for all edits have been provided as well.**

**REVIEWER 1**

General comments

This manuscript titled "African smoke particles act as cloud condensation nuclei in the wintertime tropical North Atlantic boundary layer over Barbados" by Haley M. Royer et al. is based on analyses of size-resolved particle composition (semi-quantitatively) and cloud condensation activity (quantitatively) data and dust concentration data. It concludes that African smoke is important for the atmospheric processes and cloud formation over the Caribbean. The addressed scientific question is well within the scope of ACP; the data and ideas in this manuscript are novel; and the manuscript is good structured in general.

However, the explanation of scientific methods including details in experimental settings and data analyses and uncertainties of the size-resolved chemical composition is unconvincing. In addition, the writing and interpretation especially in the Results and Discussion section are rough and need to be revised. The authors thus need to make careful revisions and corrections to improve the overall quality of the paper for publication in the journal. I would recommend the editor reconsider the manuscript only after a major revision by the authors.

Specific comments

1. Lines 90-102: whereas the possible compositional changes of smoke particles have been well explained the explanation of morphological changes is in lack. It's better to add the latter, too.

   a) The text has been edited to include information about potential morphological changes to smoke particles. See lines 107 – 110.

   "Chemical processing can also lead to morphological changes as the condensation of gaseous compounds and multiphase processes with aqueous compounds can result in the growth and sphericity of smoke particles, which in turn can affect the CCN properties of smoke particles (Abel et al., 2003; Giordano et al., 2015; Reid et al., 1998; Zhang et al., 2008)."

2. Lines 102-104: This sentence should be rewritten.

   a) The sentence has been revised to improve clarity. See lines 110 – 113.

   "The variations in the chemical and physical properties of emitted smoke particles as well as the changes these properties can undergo in transit make it difficult to predict the ability of smoke particles to act as CCN."

3. Line 185: since "EDX is considered a semiquantitative method" the uncertainty of this method in determining the size-resolved chemical composition should be elaborated because it directly influences the conclusions.

   a) CCSEM/EDX is a number-based technique in which particle size and the relative abundance for elements of interest can be accurately detected in each particle, thus it is a valid method for determining size-resolved chemistry of the aerosol loading. This method is considered semiquantitative as it can only be used to quantify the relative abundance of elements as opposed to exact concentrations. The text has been revised to make this point clearer. See lines 200 – 202.

      "CCSEM/EDX is a valid method for determining size-resolved chemistry of the aerosol loading as CCSEM excels in calculating particle size by imaging individual aerosols while EDX provides relative abundances for elements of interest (Tomlin et al., 2021)."

4. Line 188: "organic material, sea spray, dust, and anthropogenic emissions" are not parallel categories. Please revise.

   a) The sentence has been revised. See lines 204 – 205.

      "Single-particle analysis using CCSEM/EDX was limited to 16 elements found in common aerosols such as dust, sea salt, and smoke particles…"

5. Lines 192-194: "Samples collected on Si … generate a background signal as well." Does this mean except for the "carbon-coated copper grids" you also used silicon filters to collect particles for CCSEM/EDX analysis? This information should be added.

   a) Details about sampling on Si wafers have been added to the methods section. See lines 194 – 196.

      "We also collected samples on silicon wafers (Ted Pella, Inc.) which were analyzed with CCSEM/EDX to confirm the validity of carbon measurements on the carbon-coated copper grids."

6. Lines 198-201: "The algorithm operates by … and existing literature." The details in the K-means clustering analysis and the assignment of particle types should be explained at least in the supplement. How the number concentration-size distributions in Figs. 5 and S4 were determined should also be explained.

   a) Manuscript has been edited to direct readers to a more thorough explanation of the K-means clustering algorithm and particle identification assignment in the SI. See line 218 – 220.

      "A more thorough explanation of the k-means clustering algorithm and particle identification process including the plots used to perform particle identification (Figure S1) is provided in the supporting information (SI)."

The SI had been edited to include a more thorough explanation of the k-means clustering algorithm and particle identification assignment. A figure has also been included to show the plots used to do particle identification. See section "Details on the K-Means Clustering Algorithm and the Particle Identification Process" in the SI.

"Data from CCSEM/EDX analysis were imported into MatLab 2019b (MathWorks, Inc.) as a matrix of relative elemental abundances for each single particle in a sample. The data were then analyzed using k-means clustering in which groups of similar particles (clusters) are generated based on the presence and intensity of elemental peaks in the spectra of individual particles. To utilize k-means clustering, the user sets the number of clusters that the algorithm will generate. The k-means clustering algorithm will then generate clusters based on similarities between the elemental composition of individual particles. Cluster numbers were chosen based on the optimal trade-off between error minimization and chemical composition representativeness of the dataset after running the algorithm with different cluster numbers for the same sample.

To determine the identity of particles in a cluster, 4 plots are utilized that provide detailed information on size, shape and chemistry. These plots include a particle size distribution plot, a circularity plot, a weight matrix detailing the relative % of each of the 16 elements of interest in the particles, and a digital color stack plot that provides information on the distribution of elements throughout the particles in the cluster. An absolute intensity of 1% for elemental abundance presented in the weight matrix was used as a threshold value to consider an element present in a cluster. Upon identification of particle types, number fractions for each particle type can be calculated by summing together similar particle types and dividing by the total number of particles analyzed. These number fractions were generated for Fig. 4, 5, and 7. For figures 4 and 7, specific particle type number fractions were determined by dividing the number of specific particles in a particle type by the total number of particles analyzed in one day of sampling. For Fig. 5, number fractions were generated by dividing the number of particles of one particle type by the total number of particles in each size range plotted in the figure.

[Figure]

Figure S1 – Plots generated from the k-means clustering algorithm used for particle identification including a) a particle size distribution plot, b) a circularity plot, c) a weight matrix presenting relative abundance of elements in the cluster, and d) a digital color stack plot. The digital color stack plot describes the fraction of particles that contained each element of interest (height of bar) as well as the fraction of particles in the cluster in which an element made up a specific range of relative areas (size of colored bars and color of bar)."

7. In the section "Size-Resolved CCN Measurements and Data Analysis", the details in the experimental settings including flow rates of the CCNC, CPC, DMA, and SMPS should be explained.

    a) Details on the flow rates of each component in the CCNC set-up have been added to the text. See lines 227 – 229 and lines 251 – 253.

    "Flows for the size-resolved CCN set-up included a sheath:sample flow ratio of 10 for the CCN counter (sample flow rate of 0.5 L/min), a sheath flow of 8 L/min for the DMA, and a sample flow of 0.6 L/min for the CPC."

    "These values, along with the particle number size distribution determined by an SMPS (SMPS, Grimm model 5420 with CPC 3772) operating independently of the CCNC set-up, are then used to calculate the effective hygroscopicity parameter κ…"

8. Lines 243-244: "Calculation of activation … in Pöhlker et al., 2016." These should be explained at least in the supplement.

    a) We thank the reviewer for this comment. We have removed this sentence as it is extraneous and we do not describe the size-resolved CCN data or efficiencies in this work. We instead focus on CCN number concentrations and values of the

hygroscopicity parameter, κ, for which we provide the equation and an explanation of how this data is obtained.

9. Line 255: Should "Sampling Conditions" be replaced by "Air Mass Characteristics"?

   a) Text has been edited. See line 273:

   "Air Mass Characteristics during the EUREC⁴A and ATOMIC Campaigns"

10. Lines 260-267: please restructure this part. For example, only from Fig. 1, one cannot see the result described in lines 261-262. In addition, it is better to describe the figure before it is referenced.

    a) Text edited to explain that figures 1 and 2a were used to determine air mass origin. Section has been rewritten so that figures are described first. See lines 274 – 285.

    "To confirm the origins of the various air masses sampled, we performed back trajectory analysis throughout the campaign using NOAA's HYSPLIT model (Figure 1) and quantified dust mass concentrations (Figure 2a). Results of these two analyses show that Barbados was influenced by two types of air masses during the sampling period: air masses that, over the course of 6 days, do not pass over land (referred to as clean marine conditions), and air masses that have passed over the African continent (referred to as continental aerosol transport (CAT) events). Back trajectory analysis was not conducted for time periods longer than 6 days, which introduces the possibility that marine air masses could have been influenced by European outflow as well. Figure 1 shows that during periods with low dust mass concentrations and a bimodal size distribution, air masses originated from the remote Atlantic Ocean at higher latitudes with no land contact over 6 days. During time periods with high dust mass concentrations, air masses originated from continental Africa. Figure 2a shows that the total mass concentration of dust particles correlates very well with the arrival of air masses originating from Africa."

11. Lines 269-271: "suggesting that … onto larger transported continental aerosols (Tomlin et al., 2021)." This part needs to be modified. For example, "suggesting that transported particles overwhelmed the background marine particle loading so that the small Aitken mode particles possibly coagulated onto the larger transported continental particles (Tomlin et al., 2021)."

    a) The text does not attempt to suggest that transported particles overwhelming the background particle loading and Aitken mode particles are coagulating onto larger particles simultaneously, but that either occurrence may explain the formation of a unimodal size distribution. The text has been revised to clarify this point. See lines 289 – 293.

    "Upon the increase in dust mass concentrations, the submicron particle size distribution becomes unimodal and the smallest Aitken size mode is negligible, suggesting that long-range transported (LRT) particles are either dominant over

the background marine particle loading or that smaller Aitken mode particles are coagulating onto larger LRT continental aerosols to form a unimodal accumulation mode (Tomlin et al., 2021)."

12. Lines 291-346: it's better that the morphology of aged sea spray, mineral dust, and sulfate particles are also explained (and interpreted from the viewpoint of transport histories (e.g., history of atmospheric moisture conditions) if applicable).

   a) The text has been edited to include morphology of aged sea spray, mineral dust, and sulfate with references to previous literature on morphologic characteristics of each particle type. See lines 343 – 346, 352 – 354, and 368 – 369.

   "Morphologically, aged sea salt particles had either a similar appearance to fresh sea salt particles, which is often either cubic (as seen in Fig 3), or appeared as a flakey amorphous mass (Hoffman et al., 2004; Laskin et al., 2012; Li et al., 2010)."

   "Dust often appeared as a flakey or nodular amorphous mass as exhibited in Fig 3 and previous literature (Krueger et al., 2004; Laskin et al., 2005; Pachauri et al., 2013; Remoundaki et al., 2011)."

   "The morphology of sulfate particles appeared smooth and spherical as reported in previous literature (Nájera & Horn, 2009)."

13. Lines 299-305: the explanation on aged sea spray is not enough. Based on the explanation, one cannot see if the related particle and spectra presented in Fig. 3 is of aged sea spray particle or not.

   a) The text has been revised to make this particle type and its example in figure 3 clearer. See lines 337 – 340.

   "Figure 3 provides an example of an aged sea spray particle in which Na is high (indicating the presence of salt) with a low Cl peak (suggesting the particle has been aged). The presence of S in this spectrum may explain the low relative abundance of Cl compared to Na."

14. Lines 315-317: the existence of N or ammonium cannot be seen from the related particle and spectra presented in Fig. 3.

   a) Sulfate SEM/EDX image and spectra in Fig 3. have been replaced with a more exemplary particle. See line 399.

15. Lines 325-327: how about the morphologies of the smoke particles in this study, which type is dominant?

   a) Text has been edited to include dominant morphology of smoke particles. See lines 378 – 379.

"In this study, smoke particles most frequently appeared as small spherical particles."

16. Lines 330 and 334: please correct the elements of dust components.

   a) Text has been fixed. See lines 348 and 382.

17. Lines 341-342: "This scarcity of … respectively." This sentence is incomplete.

   a) The text has been revised to improve clarity. See lines 393 – 394.

      "The absence of S or N, which are often indicative of sulfate and nitrate, respectively, suggests that these particles have undergone minimal chemical aging."

18. Lines 362-364: "These finds … for smoke (Figure S2 and S3)." First, the origin of the data in Figs. S2 and S3 need to be explained. Next, can the CO variation during the observation period of this study indicate the arrival of smoke particles during the three CAT events?

   a) AOD is optical depth in the vertical direction, whereas AOT is optical thickness along an arbitrary path. We included AOT to corroborate AOD data but have since removed AOT from the manuscript to avoid confusion. The source of both AOD and CO data have been added to the figure captions of Figs. S2 and S3.

      "CO measurements were obtained from the Sentinel-5P Near Real-Time Carbon Monoxide dataset. AOD data were obtained from the Copernicus Atmosphere Monitoring Service (CAMS)."

19. Lines 378-382: please rewrite this part to make it concise.

   a) The manuscript has been revised to make the text more concise. See lines 426 – 429.

      "Figure 5 presents size-resolved chemical data from CCSEM/EDX analysis from clean marine periods (average of all clean marine periods) and one exemplary time period influenced by continental air masses (CAT event 1). Similar plots for other CAT events are provided in the SI (Figure S6)."

20. Lines 391-393: "A large decrease … observed in Figure 2." This part needs more explanations. Do the authors know any existing study that reported the condensing of marine biogenic sulfur precursors onto transported particles? Do the authors mean "The large decrease in sulfate number fraction during CAT events might be caused by the condensation of marine biogenic sulfur precursors and/or Aitken mode sulfate-containing particles onto the large, transported particles as indicated in Fig. 2."?

   a) The text has been edited to incorporate the suggested change. See lines 441 – 444.

> "The large decrease in sulfate number fraction during CAT events might be caused by the condensation of Aitken mode sulfate-containing particles onto larger, transported particles as indicated in Fig. 2 (Gaston et al., 2010)."

21. Lines 399-401: please explain "Bin sizes for each decade" and "bin size"; please rewrite this sentence.

   a) Text has been edited to clarify definition of "bin'. See lines 450 – 453.

   > "Particles were organized into 32 size cuts (bins) to maximize resolution of size-resolved chemical data. Particle counts in each bin range from 34 particles to up to 3041 with an average bin size of 493 particles for the Clean Marine plot and 973 for the CAT Event plot."

22. Line 410: why these small particles are less hygroscopic should be tentatively explained.

   a) The text has been edited to explain the low hygroscopicity of smaller particles. See lines 460 – 464.

   > "Then, with each subsequent increase in S after 0.24% S, $\kappa$ decreases as smaller, less hygroscopic particles activate at higher supersaturations. The low hygroscopicity of these smaller particles can be explained by compositional changes in the aerosol loading exhibited in Fig 5, reflecting the shift in particle chemistry from mostly sulfate to mostly organic with decreasing particle size."

23. Lines 414-423: the difference between marine conditions and smoky conditions should be quantitatively explained and the data should be tabled for all S conditions.

   a) Marine conditions and conditions influenced by continental aerosols are distinguished by dust mass concentrations and HYSPLIT back trajectories as mentioned in section "Sampling Conditions during the EUREC[4]A and ATOMIC Campaigns" and the caption for figure 1. Quantitative differences between clean marine and continentally influenced conditions as well as the time period for each condition throughout the campaign are provided in Table 1. The text has been edited to make this clearer. See lines 515 – 516.

   > "Table 1 provides averages of CCN concentrations for each time period shown in Fig. 7b."

24. Lines 418-420: the first half and the second half of the sentence are repeating each other.

   a) The text has been revised to be more concise. See lines 472 – 473.

   > "As expected, trends in average $d_{50}$ for both plots indicate that smaller particles activate as CCN with larger supersaturations."

25. Lines 428-432: "However, one finding of note … African aerosols." Please rewrite this part to make it concise.

a) The manuscript has been revised to make the text more concise. See lines 482 – 485.

"However, Good et al., (2010) shows similarities to our work through the distinct drop in κ between clean marine conditions and CAT conditions which is attributed to the addition of hydrophobic dust to the aerosol loading."

26. Line 431: "such as dust that activate as CCN". Dust might haven't acted as CCN but it still could have influenced the activation diameter and thus the overall estimated kappa. I suggest deleting "that activate as CCN".

    a) The phrase "that activate as CCN" has been removed from this sentence. See line 482 – 485.

    "However, Good et al., (2010) shows similarities to our work through the distinct drop in κ between clean marine conditions and CAT conditions which is attributed to the addition of hydrophobic dust to the aerosol loading."

27. Lines 441-443: the existence of organics may have lowered the observed kappa. Please rethink this conclusion.

    a) The text has been revised. See lines 494 – 500.

    "They conclude that the low κ values observed during their sampling were due to organic compounds activating as CCN. We find similar κ values to Kristensen et al. (2016) and a similar particle chemistry of the accumulation and Aikten modes during clean marine conditions, suggesting that organics and sulfates were the primary CCN types during clean marine conditions studied for the EUREC[4]A and ATOMIC campaigns as well. We also observe a drop κ from clean marine conditions to CAT events. This drop in κ indicates the influence of an additional CCN particle type contributed by the CAT events."

28. Lines 444-447: please explain the box plots in detail and the meaning of the whiskers of $d_{50}$.

    a) The caption for figure 6 has been revised to provide further detail about the plots. See lines 502 – 508.

    "Figure 6: Hygroscopicity parameter κ (left axis, box plots) and corresponding mean diameter at which 50% of the particles in an aerosol population activate as CCN at a particular S, also called the critical diameter "d50" (right axis; red markers) for the investigated supersaturations (S). Whiskers on "d50" markers represent standard deviation values of "d50". Black dots in the boxplot indicate κ mean values. Boxes represent the upper quartile, median, and lower quartile κ values at each S. Whiskers represent the upper and lower limit of κ at each S."

29. Lines 458-459: "In Fig 7b, there is a clear and direct relationship" should be rewritten. For example, "Figure 7b shows a positive correlation". In addition, this positive correlation should be further elaborated with the data for example in Table 1.

    a) Text has been edited to include the suggested changes as well as a note referring to the SI for an additional figure supporting the relationship between smoke number fraction and CCN concentrations. See lines 522 – 524.

    "Figure 7b shows a positive correlation between smoke number fractions and CCN counts. A correlation plot of smoke number fraction and CCN concentrations is also provided in Fig. S7 to further emphasize their direct relationship."

30. Line 460: "may act as CCN" should be "may have acted as CCN".

    a) Text has been revised. See lines 525 – 526.

31. Lines 464-466: "The large number of smoke particles … on sulfate or organic particles." Do you want to say that the number of sulfate and organic particles were not enough for the measured CCN counts?

    a) This sentence has been removed from the manuscript.

32. Lines 469-472: what are the possible surfactants? Could you explain?

    a) The referenced text has been removed from the manuscript as it is not pertinent to the study described.

33. Lines 472-479: First, can any evidence be given from the single particle composition data that aging of particles occurred. Second, what you want to conclude from this part is not clear.

    a) The referenced text has been removed from the manuscript as it is not pertinent to the study described.

34. Lines 485-486: "It is likely that … of factors." seems redundant. It's better to delete it.

    a) The text has been revised.

35. Lines 450-488: the writing and organization of this paragraph are quite rough and should be revised.

    a) The text has been revised to be more concise. See lines 511 – 537.

36. Table 1 should be modified according to https://www.atmospheric-chemistry-and-physics.net/submission.html#figurestables

    a) Table 1 has been reformatted. See line 547.

37. Line 508: the activation diameter should be explained with the measured S condition.

    a) The text has been revised to include the S at which these measurements were obtained. See line 556 – 558.

"Upon the arrival of African aerosols to BACO, CCN average activation diameter increased to approximately 200 nm while the average hygroscopicity of activated particles for all CAT events decreased to $\kappa = 0.45+0.1$ for 0.16% S."

38. Line 513: "Overall, we find that smoke has a larger effect on CCN number concentrations than dust." Does this conclusion apply for all time or for the studied period or season?

    a) The influence of smoke being greater than dust likely applies only to the boreal winter/spring when smoke transport is high due to the occurrence of the Sahelian burn season. The text has been edited to make this point clear. See lines 562 – 563.

    Overall, we find that smoke has a large effect on CCN number concentrations during the boreal winter when smoke transport is high.

39. Line 522: what's the difference between AOD and AOT? And where are the data of Figs. S2 and S3 from?

    a) AOD is optical depth in the vertical direction, whereas AOT is optical thickness along an arbitrary path. We included AOT to corroborate AOD data, but have since removed AOT from the manuscript to avoid confusion. The source of both AOD and CO data have been added to the figure captions of Figs. S2 and S3.

    "CO measurements were obtained from the Sentinel-5P Near Real-Time Carbon Monoxide dataset. AOD data were obtained from the Copernicus Atmosphere Monitoring Service (CAMS)."

Technical corrections

1. Line 73: Should "though" be replaced by "however" or other similar phrases?

    a) Text has been revised. "Ultimately, though…" has been replaced with "However…". See line 75.

2. Line 143: "composition" should be "compositions".

    a) Text has been edited. See line 153.

3. Line 182: "affects" should be "affect".

    a) Text has been edited. See line 198.

4. Line 218: "Rose et al., 2008" should be "Rose et al. (2008)". Please modify similar expressions in other places.

    a) The suggested changes have been made throughout the text, where applicable.

5. Line 227: "0.71" should be "0.71 %".

    a) Text has been edited. See line 246.

6. Lines 251-252: "single particle data (e.g., CCSEM-EDX), and air mass history (e.g., NOAA's HYSPLIT model)", the "e.g.," in the parentheses should be deleted.

   a) Text has been edited. See line 270.

7. The color shadings in Fig. 2 should have been explained in the caption.

   a) Text has been edited to explain color shading. See lines 310 – 312.

      "Color shading in (a) represents continental aerosol influence (orange shading) and clean marine influence (b) as determined by NOAA HYSPLIT back trajectories calculated at Ragged Point."

8. Line 350: "represent" should be "present"?

   a) Text has been edited. See line 406.

9. Line 362: "observe" should be "observed".

   a) Text has been edited. See line 416.

10. Lines 369-370: "wildfire smoke appears to overwhelm the number fraction of the submicron aerosol loading" should be "wildfire smoke particles appeared to dominate the number fraction of submicron aerosol." Please also correct the wrong use of overwhelm in other places throughout the manuscript, e.g., line 510.

    a) Text has been edited on the specific line referenced as per the suggestion and throughout the text by replacing "overwhelm" with "dominate"

11. Line 388: "Figure 4" should be "Figure 5".

    a) Text has been edited. See line 438.

12. Line 452: "N Atlantic MBL" should be "North Atlantic MBL".

    a) Text has been edited. See line 513.

13. Line 455: "Fig 7a suggests" should be "Figure 7 suggests". Please refer to https://www.atmospheric-chemistry-and-physics.net/submission.html#figurestables

    a) Text has been edited. See line 518.

14. Line 476: "to oxidants" should be "to oxidize".

    a) The referenced text has been removed from the manuscript as it is not pertinent to the present study.

**REVIEWER 2 COMMENTS**

This work reports size-resolved single particle composition and cloud condensation activity results during the EURECA and ATOMIC campaigns from January to February 2020 over the tropical North Atlantic. They concluded that aerosol hygroscopicity and CCN number concentrations during wintertime long-range transport events over the Caribbean are affected by African smoke more than dust. The main content of this manuscript is well within the scope of ACP. The manuscript is well-prepared in general. However, as the other referee commented, the writing and interpretation of your results, especially the part to support of your major conclusion, need to be carefully revised. I would also recommend the publish of this manuscript after the following issues are fully resolved.

General comments:

1.  A major part of your results is size-resolved chemical composition, which you used to support your main conclusion. However, from your methods, I did not see a clear explanation how you obtained this "size-resolved" information and it should be an really important part for your methodology. As you stated "EDX is a semiquantitative method", then how did you ensure the accuracy of the results in Fig. 4, Fig.5 and Fig. 7, and eventually drove your conclusions?

    a.  The text has been revised both in the manuscript and in the supplementary information document to explain how CCSEM/EDX is a valid method for determining size-resolved chemistry as well as the method for particle identification using CCSEM/EDX products. See lines 200 – 202.

        "CCSEM/EDX is a valid method for determining size-resolved chemistry of the aerosol loading as CCSEM excels in calculating particle size by imaging individual aerosols while EDX provides the relative abundances for elements of interest (Tomlin et al., 2021)."

        Manuscript has been edited to direct readers to a more thorough explanation of the K-means clustering algorithm and particle identification assignment in the SI. See lines 218 – 220.

        "A more thorough explanation of the k-means clustering algorithm and particle identification process including the plots used to perform particle identification (Figure S1) is provided in the Supporting Information (SI)."

        The SI had been edited to include a more thorough explanation of the k-means clustering algorithm and particle identification assignment. A figure has also been included to show the plots used to do particle identification.

        "Data from CCSEM/EDX analysis were imported into MatLab 2019b (MathWorks, Inc.) as a matrix of relative elemental abundances for each single particle in a sample. The data were then analyzed using k-means clustering in

which groups of similar particles (clusters) are generated based on the presence and intensity of elemental peaks in the spectra of individual particles. To utilize k-means clustering, the user sets the number of clusters that the algorithm will generate. The k-means clustering algorithm will then generate clusters based on similarities between the elemental composition of individual particles. Cluster numbers were chosen based on the optimal trade-off between error minimization and chemical composition representativeness of the dataset after running the algorithm with different cluster numbers for the same sample.

To determine the identity of particles in a cluster, 4 plots are utilized that provide detailed information on size, shape and chemistry. These plots include a particle size distribution plot, a circularity plot, a weight matrix detailing the relative % of each of the 16 elements of interest in the particles, and a digital color stack plot that provides information on the distribution of elements throughout the particles in the cluster. An absolute intensity of 1% for elemental abundance presented in the weight matrix was used as a threshold value to consider an element present in a cluster. Upon identification of particle types, number fractions for each particle type can be calculated by summing together similar particle types and dividing by the total number of particles analyzed. These number fractions were generated for Fig. 4, 5, and 7. For figures 4 and 7, specific particle type number fractions were determined by dividing the number of specific particles in a particle type by the total number of particles analyzed in one day of sampling. For Fig. 5, number fractions were generated by dividing the number of particles of one particle type by the total number of particles in each size range plotted in the figure.

[Figure]

Figure S1 – Plots generated from the k-means clustering algorithm used for particle identification including a) a particle size distribution plot, b) a circularity plot, c) a weight matrix presenting relative abundance of elements in the cluster, and d) a digital color stack plot. The digital color stack plot describes the fraction of particles that contained each element of interest (height of bar) as well as the

fraction of particles in the cluster in which an element made up a specific range of relative areas (size of colored bars and color of bar)."

2. According to Fig.7b, you stated the CCN counts correlate quite well with smoke number fraction. Could you also plot similar figure for dust number fraction with CCN counts?

   a. A correlation plot for smoke number fractions and submicron CCN concentrations has been added to the SI. Because few dust particles were observed in the submicron aerosol fraction, a similar correlation plot with dust number concentrations was not added. The manuscript has been revised as well to direct readers to the SI as well. See lines 523 – 524.

   "A correlation plot of smoke number fraction and CCN concentrations is also provided in Fig. S7 to further emphasize their direct relationship."

3. Continued with my comment above, you only measured the chemical composition of particles at diameter < 0.7μm. You also agreed that there might be more dust particles in the supermicron range. These particles may also be good CCN as they are large enough. However, you did not measure those. From this perspective, I might not agree that smoke has a larger effect on CCN number concentration than dust. You should consider how to re-interpret your results.

   a. Analysis of supermicron particles from stages 1 and 2 of the MPS were conducted with CCSEM/EDX but were not presented in this paper as it was not relevant to the submicron CCN data shown. The manuscript has been updated to notify the reader that both supermicron and submicron analysis was performed. See lines 409 – 411.

   "A similar plot to Fig. 4 that contains temporal chemistry from stage 1 and stage 2 of the MPS (representing supermicron particles >0.7 μm diameter) determined using CCSEM/EDX analysis can be found in the SI (Figure S2)."

   The results section has been updated (including Fig. 3) to include particle types not present in the submicron particle loading but present in the supermicron particle loading. See lines 324 – 333; lines 355 – 362;

   *"Sea Spray*
   Sea spray particles were characterized by high relative abundance of approximately equal parts Na and Cl, indicating the formation of halite (NaCl). Morphologically, sea spray particles have a cubic shape that represents the crystal structure of halite. Small Mg peaks approximately 10% of the height of Na peaks were also observed in NaCl particles and reflect the Na:Mg ratio of seawater. Additional components of sea spray particles include rod-shaped particles containing Ca and S (presumably calcium sulfate) that were often found attached to NaCl particles (Ault et al., 2013; Bondy et al., 2018; Choël et al., 2007). Elements such as N and S that may suggest aging of sea spray were either absent

or present in small relative abundance on NaCl components of sea spray particles."

*"Internally Mixed Mineral Dust and Sea Spray*
Particles containing elements indicative of both mineral dust (Si, Al, Fe, K, Ca, and Mg) and sea salt (approximately equal relative abundances of Na and Cl) were characterized as internally mixed mineral dust and sea spray (Choël et al., 2007; Deboudt et al., 2010; Sobanska et al., 2014). Elements such as S and N were often not present in this particle type, suggesting the particles had not undergone atmospheric aging during transit. Particles containing both dust and sea spray components often appeared as conglomerates of multiple particles with some parts containing more sea spray components and others containing more mineral dust components."

The SI has also been revised to include a similar plot to Figure 4 which includes data from all MPS stages to showcase the particle chemistry of the total aerosol loading throughout the entire sampling period.

[Figure]

"Figure S2 - Temporal evolution of particle type number fractions for a) stage 1 and b) stage 2 of the multistage particle sampler analyzed using CCSEM/EDX."

4. On page 24, you gave several reasons that smoke particles may act as CCN. However, I did not really agree with those reasons you provided starting from line 466 til the end of this paragraph. First, you did not have any direct evidences for these explanations. Second but more importantly, you just mentioned in line 405 that smoke particles lower submicron aerosol hygroscopicity. Then these particles can act as CCN would not be due to the elevated hygroscopicity or κ, which arises from the presence of water-soluble compounds, for instance WOSC, aged organics or other salts.

a. The referenced text has been removed from the manuscript as it is not pertinent to the study discussed.

Specific comments:

1. Line 161-162: BC particles did not evaporate at 500 °C, either. How did you exclude them?

   a. It is possible that smoke mass is included in dust mass concentrations. However, smoke mass would be negligible compared to dust mass if it is included. This method has been verified and has been used for calculating dust mass concentrations for decades. The text has been revised to describe the validity of this method. See lines 173 – 176.

   "Previous research has confirmed the validity of this method for determining dust mass concentrations through chemical analysis of dust ash determined from filters collected in Barbados, crustal abundance, and soil dust composition (Zuidema et al., 2019)."

2. Line 185-203: Please provide detailed information how you obtained size-resolved chemical composition for particles.

   a. The manuscript has been revised to direct the reader to the SI for further information on determining size-resolved chemical composition. See lines 218 – 220.

   "A more thorough explanation of the k-means clustering algorithm and particle identification process including the plots used to perform particle identification (Figure S1) is provided in the Supporting Information (SI)."

   A section has been added to the SI to describe the method for determination of size-resolved chemistry and particle identification using CCSEM/EDX.

   "Data from CCSEM/EDX analysis were imported into MatLab 2019b (MathWorks, Inc.) as a matrix of relative elemental abundances for each single particle in a sample. The data were then analyzed using k-means clustering in which groups of similar particles (clusters) are generated based on the presence and intensity of elemental peaks in the spectra of individual particles. To utilize k-means clustering, the user sets the number of clusters that the algorithm will generate. The k-means clustering algorithm will then generate clusters based on similarities between the elemental composition of individual particles. Cluster numbers were chosen based on the optimal trade-off between error minimization and chemical composition representativeness of the dataset after running the algorithm with different cluster numbers for the same sample.

   To determine the identity of particles in a cluster, 4 plots are utilized that provide detailed information on size, shape and chemistry. These plots include a particle

size distribution plot, a circularity plot, a weight matrix detailing the relative % of each of the 16 elements of interest in the particles, and a digital color stack plot that provides information on the distribution of elements throughout the particles in the cluster. An absolute intensity of 1% for elemental abundance presented in the weight matrix was used as a threshold value to consider an element present in a cluster. Upon identification of particle types, number fractions for each particle type can be calculated by summing together similar particle types and dividing by the total number of particles analyzed. These number fractions were generated for Fig. 4, 5, and 7. For figures 4 and 7, specific particle type number fractions were determined by dividing the number of specific particles in a particle type by the total number of particles analyzed in one day of sampling. For Fig. 5, number fractions were generated by dividing the number of particles of one particle type by the total number of particles in each size range plotted in the figure.

[Figure]

Figure S1 – Plots generated from the k-means clustering algorithm used for particle identification including a) a particle size distribution plot, b) a circularity plot, c) a weight matrix presenting relative abundance of elements in the cluster, and d) a digital color stack plot. The digital color stack plot describes the fraction of particles that contained each element of interest (height of bar) as well as the fraction of particles in the cluster in which an element made up a specific range of relative areas (size of colored bars and color of bar)."

3. Line 202: Similar to my general comments, since you studied particles at size below 0.7µm, where dust might not be as abundant as smoke, then how did you drive the conclusion about the dust effect on CCN activity?

    a. This sentence has been removed from the manuscript.

4. Line 246-248: How did you prove current statement? From which figure? This sentence fits better into conclusion part, but not the starting of results. Again, how did you identify smoke particles? It came up without any description, though you explained it later afterwards. Please consider rephrase your results structure.

    a. The sentence has been revised. See lines 264 – 266.

       "In this section we will show that, upon arrival of co-transported dust and smoke, smoke originating from fires in the African Sahel dominate the accumulation mode particle population in the tropical North Atlantic MBL, which results in an increase in CCN number concentration."

5. Line 266: Were the Aitken size mode particles from the marine background? Why?

    a. The bimodal size distribution has been shown to be characteristic of marine background conditions as shown in previous literature. Previous studies have shown that in clean marine conditions the accumulation mode is primarily composed of sulfates while the Aitken mode is primarily composed of organics. The text has been revised to include these details. See lines 287 – 289.

       "During time periods when dust concentrations were low, the particle loading has a bimodal size distribution characteristic of clean marine air masses (Figure 2b; Ault et al., 2013; Hoppel et al., 1986; O'Dowd et al., 2004)."

6. Line 315 and 317: strong of what? Please rephrase it.

    a. The text has been edited to make the language clearer. See lines 364 – 368.

       "Sulfate-rich particles are a prevalent component of marine submicron aerosol (O'Dowd & de Leeuw, 2007) and characterized here by a dominant S component - often with high relative abundance of C, O, and N… The high relative abundance of C indicates a large organic fraction as well."

7. Line 325 to 327: Then what is the morphology of smoke particles in your study? You should provide more information of your current results.

    a. The text has been edited to note the dominant morphology of smoke particles. See lines 378 – 379.

       "In this study, smoke particles most frequently appeared as small spherical particles."

8. Line 354: How did you obtain the number fractions of each kind of particles? Please add this part into your methodology.

    a. Details on the calculation of number fractions for each particle type have been added to the SI.

"Upon identification of particle types, number fractions for each particle type can be calculated by summing together similar particle types and dividing by the total number of particles analyzed. These number fractions were generated for Fig. 4, 5, and 7. For figures 4 and 7, specific particle type number fractions were determined by dividing the number of specific particles in a particle type by the total number of particles analyzed in one day of sampling. For Fig. 5, number fractions were generated by dividing the number of particles of one particle type by the total number of particles in each size range plotted in the figure."

9. Line 366 to 370: Please consider rewrite the sentence, it is confusing.
   a. This sentence has been revised to improve clarity. See lines 420 – 421.

   "Upon arrival of continental aerosols, wildfire smoke appeared to dominate the number fraction of submicron aerosol."

10. Line 381: From Fig. 5, I cannot directly see those particles are the largest ones. Maybe you can plot a figure for the normalized mean diameter of each particle type, which is more obvious.

    a. A table has been added to the SI providing average diameters of each particle type during each sampling period. The text has also been revised to direct the reader to the SI table. See lines 429 – 430.

    "Average particle diameters for each particle type detected during each sampling period is also provided in Table S1."

11. Line 382-384: I cannot understand what do you mean "clean" here, please clarify. The number concentration or mass concentration of particles were lower for clean marine conditions?

    a. True clean marine conditions would be completely devoid of any continental aerosols such as dust and smoke. However, in our data we see that dust and smoke are still present during time periods when dust mass concentrations are minimal and HYSPLIT trajectories indicate air mass origins from the Atlantic Ocean (as opposed to continental Africa). Thus, our "clean marine conditions" are not, by the strictest definition "clean marine" but rather "clean" in relation to time periods influenced by high concentrations of dust and smoke. The text has been edited to make this point clearer. See lines 431 – 433.

    "This suggests that our 'clean marine conditions' are only 'clean' relative to time periods influenced by dust and smoke, rather than pristine clean marine conditions without any continental aerosol influence."

12. Line 408: Sometimes you used "dusty conditions", sometimes you used "continental influenced aerosols" or sometimes referred to "CAT Event". At line 415, you used "smokey conditions". Could you please be consistent?

    a. The text has been revised throughout to use "CAT conditions". The introduction section has been revised to notify the reader of the meaning for CAT conditions to make it clear early-on what its meaning is. See lines 114 – 117.

    "In this study, we investigated the relationship between submicron aerosol composition and CCN in the tropical north Atlantic MBL during marine background conditions and conditions affected by long-range continental aerosol transport of smoke and dust particles (henceforth referred to as "CAT" conditions)."

13. Line 427 to 428: Did they obtain those values at the same S? Otherwise, you cannot directly compare them.

    a. Good et al. (2010) used similar S values as those we used in our work.
        i. Good et al., 2010 – 0.11, 0.18, 0.34, 0.5, and 0.75% S
        ii. Royer et al., 2022 – 0.09, 0.16, 0.24, 0.43, and 0.74% S

14. Line 454: Could you compare the results at those two CAT Event or for those three Clean Marine Period? For instance, for those two CAT events, the CCN concentration was different with different hygroscopicity, even though they are both influenced by the continental transport. Could you please give the potential reasons and make proper discussions.

    a. A table has been added to the SI containing average and median values of smoke particle numbers analyzed for each CAT Event. The manuscript has been revised to direct the reader to this table and explain the use of this data. See lines 516 – 518.

    "Table S2 also provides average and median counts of analyzed smoke particles calculated for each CAT event, including CAT Event 3 which had fewer smoke particles and a higher $\kappa$ compared to other CAT Events."